# Analysis of Plantar Pressure Pattern after Metatarsal Head Resection. Can Plantar Pressure Predict Diabetic Foot Reulceration?

**DOI:** 10.3390/jcm10112260

**Published:** 2021-05-24

**Authors:** Marta García-Madrid, Yolanda García-Álvarez, Francisco Javier Álvaro-Afonso, Esther García-Morales, Aroa Tardáguila-García, José Luis Lázaro-Martínez

**Affiliations:** Diabetic Foot Unit, Clínica Universitaria de Podología, Facultad de Enfermería, Fisioterapia y Podología, Universidad Complutense, Instituto de Investigación Sanitaria del Hospital Clínico San Carlos (IdISSC), 28040 Madrid, Spain; magarc28@ucm.es (M.G.-M.); alvaro@ucm.es (F.J.Á.-A.); eagarcia@ucm.es (E.G.-M.); aroa_tg@hotmail.com (A.T.-G.); diabetes@ucm.es (J.L.L.-M.)

**Keywords:** diabetic foot, conservative surgery, metatarsal head resection, pressure transfer, reulceration

## Abstract

To evaluate the metatarsal head that was associated with the highest plantar pressure after metatarsal head resection (MHR) and the relations with reulceration at one year, a prospective was conducted with a total of sixty-five patients with diabetes who suffered from the first MHR and with an inactive ulcer at the moment of inclusion. Peak plantar pressure and pressure time integral were recorded at five specific locations in the forefoot: first, second, third, fourth, and fifth metatarsal heads. The highest value of the four remaining metatarsals was selected. After resection of the first metatarsal head, there is a displacement of the pressure beneath the second metatarsal head (*p* < 0.001). Following the resection of the minor metatarsal bones, there was a medial displacement of the plantar pressure. In this way, plantar pressure was displaced under the first metatarsal head following resection of the second or third head (*p* = 0.001) and under the central heads after resection of the fourth or fifth metatarsal head (*p* < 0.009 and *p* < 0.001 respectively). During the one-year follow-up, patients who underwent a metatarsal head resection in the first and second metatarsal heads suffered transfer lesion in the location with the highest pressure. Patients who underwent a minor metatarsal head resection (second–fifth metatarsal heads) showed a medial transference of pressure. Additionally, following the resection of the first metatarsal head there was a transference of pressure beneath the second metatarsal head. Increase of pressure was found to be a predictor of reulceration in cases of resection of the first and second metatarsal heads.

## 1. Introduction

Diabetic foot ulcer (DFU) is one of the most frequent complications associated with diabetes mellitus. The incidence of diabetic foot ulcers is approximately 19–34%, and more than 50% of these DFU will become infected [1]. It is estimated that around 20 to 60% of diabetic foot infections will involve the bone [2] resulting in different levels of amputation [1]. The forefoot is the main location of DFU occurrence especially the plantar surface of the metatarsal heads due to the presence of diabetic polyneuropathy (DPN), deformities, and previous forefoot surgery [3]. Patients with DM and DPN have shown an increase of peak plantar pressure (PPP) compared to healthy subjects and diabetes controls without DPN [4].There is an association between the highest PPP in neuropathic patients and previous history of DFU [5], deformities [6], and minor amputation [7]. Additionally, the forefoot area displays an increase in these parameters versus the rest of the foot in these patients [5]. The International Working Group on the Diabetic Foot (IWGDF) recommend analysis of plantar pressure in patients with diabetes and at a high risk for ulcer occurrence (IWGDF, risk 3) [8]. Dynamic barefoot pressure measurement can predict the areas at risk of developing a DFU; it has been extensively studied in recent years [9,10].

Metatarsal head resection (MHR) is a common and safe procedure used as a prophylactic and curative technique for DFU treatment [11,12]. MHR reduces plantar pressure at the ulcer site. However, MHR also leads to biomechanical changes leading to short- and long-term complications [13]. Reulceration is the main complication associated with MHR due to the transfer of plantar pressure to the adjacent metatarsal joints and consequently the occurrence of a new ulcer. Reulceration is seen in approximately 40% of the patients within 1 year after ulcer healing [1]. Previous studies showed an increase in PPP in the adjacent metatarsal regions after MHR [14]. The literature showed a transference of pressure to the mid-region of the foot after the resection of the second and third metatarsal heads [14] and a transferred pressure to the second, third, and fourth metatarsal heads after a first ray amputation [15]. Recent studies have highlighted the importance of classifying patterns of forefoot loading in diabetic patients [16]. This could be helpful to finding a pressure transference pattern in patients with a history of MHR to implement preventive strategies such as orthotic interventions or prophylactic surgeries. There are few studies linking the increase in plantar pressure with the location of the DFU after diabetic foot surgery, and the results are inconsistent [14,15]. It could be useful to determining a plantar pressure pattern distribution after forefoot surgery procedures. 

Therefore, the principal aim of our study was to evaluate the metatarsal head that was associated with the highest PPP and PTI after MHR and the relations with reulceration at one year.

## 2. Materials and Methods

### 2.1. Subjects

A prospective study was conducted between May 2018 to June 2020 in 65 diabetic patients in a specialized diabetic foot unit. 

The target sample size was calculated using Epidat^®^ v.4.2 for Mac OS. It was determined that the standard deviation was 5.4 to detect a difference of at least two times for peak plantar pressure changes after metatarsal head resection [14], based on a desired power of 80% with a β level of 20%, α level of 0.05, and confidence interval of 95%. Assuming a loss of 0.1% due to observational study design, at least 64 participants must be included in the study.

The inclusion criteria were as follows: patients with diabetes type 1 or 2 aged > 18 years old, patients without active foot ulcer at inclusion, and patients who underwent their first metatarsal head resection surgery due a previous forefoot plantar ulcer. 

The exclusion criteria were as follows: need for walking aids, Charcot neuroarthropathy, presence of critical limb ischemia, amputation of the contralateral limb, and inability to walk autonomously.

### 2.2. Clinical Evaluation

Neuropathy was confirmed by the loss of sensation of three plantar foot sites with a 10-g Semmes–Weinstein monofilament and/or vibration perception threshold >25 mV as assessed with the biothensiometer (Me.Te.Da. s.r.l., Via Silvio Pellico, 4, 63074 San Benedetto del Tronto AP, Italy) [17].

Peripheral arterial disease (PAD) was defined as the absence of both distal pulses and/or an ankle brachial index (ABI) of <0.9. In patients whose ABI was >1.4 or those with uncertain diagnostic findings, a systolic toe pressure of <55 mmHg, systolic ankle pressure <70 mmHg, or a toe brachial index (TBI) < 0.7 confirmed the diagnosis of PAD. Critical limb ischemia was diagnosed as the absence of both distal pulses and an ABI of <0.39, ankle systolic pressure <50 mmHg, and toe pressure <30 mmHg [18].

All patients underwent an MHR as a prophylactic procedure due to the presence of deformity without the possibility of being offloaded; they could also have a curative MHR for the clinical suspicion of osteomyelitis [11]. All patients received standard of care consisting of ulcer debridement and proper offloading following the IWGDF guideline recommendations until healing. 

Healing was defined as presence of intact skin without any drainage of a previous foot ulcer site [19]. After 1 month of confirmed healing, patients were screened for biomechanical and plantar pressure distribution.

### 2.3. Biomechanical and Plantar Pressure Assessment

An experienced podiatrist (MGM) identified biomechanical characteristics and deformities. Forefoot deformities were evaluated with the patient in the standing position. We considered the presence of hallux abductus valgus (HAV), hammer toe, Taylor bunion, and plantar prominence of the metatarsal head [20].

A dynamic pressure measurement system, FootScan^®^ software (RScan International, 3583 Olen, Belgium) was used to record PPP (N/cm^2^) and PTI (N/cm^2^/s). This process used two-meter-long platforms with 4 sensors/cm^2^ and a 3D-Box interface that was synchronized with the motion capture system. Patients were instructed to walk barefoot over three minutes before recovering the plantar pressure to accommodate the patient to normal gait and speed. After this, four registers were taken to calculate the mean of both measurements (PPP and PTI) with a two-step approach to the platform [21]. The PPP and PTI were recorded at five specified locations in the forefoot: first, second, third, fourth, and fifth metatarsal head. The region corresponding with the metatarsal head resected was not selected for the analyses. The highest value of the four remaining metatarsals was selected.

After biomechanical assessment, the same experienced podiatrist prescribed a custom-made insole. A positive plaster cast of the foot was created from a static foam box impression of the foot under semi-weight bearing position. The metatarsal region on the plaster cast was marked to guide the offloading (cut-out and metatarsal bar). The insole consisted of a 5 mm micro-cork base added to a 6 mm mid-layer of ethylene vinyl acetate (EVA, shore 40A). Apex height of the metatarsal bar was 8–10 mm. Additionally, in the areas identified at risk because of the highest PPP and PTI, 5 mm EVA was removed and padded with 3 mm Plastazote^®^, shore 25A. Finally, the insoles were finished with a top cover of 3-mm-thick PPT^®^ [22]. In addition, the subject was fitted with an extra-depth therapeutic shoe consisting of a rigid rocker outsole with a pivot point at 60 percent of shoe length, a rocker angle of 20 degrees, and an upper of stretch material (Podartis, Montebelluna, Italy) [23].

All patients were followed-up for one year to record complications. Patients came monthly to the outpatient clinic to perform debridement of high-risk points, such as minor lesions according to IWGDF guidelines [24].

Reulceration was defined as the development of a transfer lesion that appeared under the head of a metatarsal other than the one previously treated [25]. Reulceration was evaluated by a different clinician to avoid bias between pressure analyses and ulcer identification. 

### 2.4. Outcome Measures

The main outcome of the study was to evaluate PPP and PTI after first metatarsal head resection surgery in the remaining metatarsal heads.

The secondary outcome of the study was to assess the correlation between the highest PPP and PTI after metatarsal head resection and its relationship with reulceration during the one-year follow-up period.

### 2.5. Statistical Analysis

The assumption of normality of all continuous variables was verified using the Kolmogorov-Smirnov test. Normally distributed variables (Kolmogorov-Smirnov test with *p* ≥ 0 05) were reported as mean and standard deviations, and non-normally distributed variables (Kolmogorov-Smirnov test with *p* < 0.05) were reported as medians and interquartile ranges. A Friedman test for paired samples was used to explore the metatarsal that supports the highest plantar pressure after MHR because of the non-normal distribution of the variable. A Wilcoxon-Mann Whitney for independent samples was used to explore the relation among the highest PPP and PTI with reulceration. 

All statistical analyses were performed using SPSS statistics version 25.0 for Mac OS (SPSS, Chicago, IL, USA). *p* values < 0.05 were considered statistically significant with a confidence interval of 95%.

## 3. Results

A total of 230 patients were evaluated. Figure 1 shows the flow chart of the 65 participants. No patient was evaluated for both feet. Demographic and foot characteristics are shown in Table 1. 

### 3.1. Main Outcome

Following the MHR of the 1st head, 60% of the patients had the highest PPP under the second metatarsal head; 62.5% of the patients who undergo a MHR of the second metatarsal had the highest PPP beneath the first metatarsal; and 70% of those who suffered a MHR of the third had the highest PPP under the first. Finally, after resection of the fourth and fifth MH, 75% and 60% of the patients displayed the highest PPP under the second and third metatarsal heads, respectively. PTI showed the same trend as PPP. Statistical differences are shown in Table 2 and Table 3, respectively. 

### 3.2. Secondary Outcomes

The one-year follow-up showed that the highest PPP and PTI under the second metatarsal after an MHR procedure of the first metatarsal was related with the development of a transfer lesion (*p* = 0.006 and *p* = 0.005, respectively). Furthermore, the maximum PPP and PTI were observed under the first MH after resection of the second was related with reulceration (*p* = 0.017 and *p* = 0.013, respectively). Despite this, no association was observed between the highest PPP and PTI patterns after resection of the third, fourth, and fifth metatarsal heads with reulceration in the one-year follow-up period. 

## 4. Discussion

The results showed that there is a specific plantar pressure pattern distribution after the first metatarsal head resection procedure. There is a displacement of the plantar pressure beneath the second metatarsal head after resection of the first metatarsal head. There was a medial displacement of the plantar pressure following the resection of the minor metatarsal bones. In this way, the highest plantar pressure was located under the central heads after resection of the fourth or fifth metatarsal head. Plantar pressure was displaced under the first metatarsal head following resection of the second or third. 

Plantar pressure was a predictor of reulceration in cases of MHR of the first and second metatarsal heads. However, the increase in plantar pressure was not always associated with reulceration. The resection of the first metatarsal head breaks the integrity of the medial column producing a shortening of this segment of the foot. Previous studies have shown a transfer of hyperloading to the adjacent rays induced by surgical intervention [26]. This proceeds via transfer of pressure from the first to the second metatarsal head. 

On the other hand, the medial column formed by the first, second, and third ray represents the longest metatarsal bones. Previous research has demonstrated that the longest metatarsal bone results in the highest plantar pressure [27]. This could explain the medial displacement seen after resection of the minor metatarsal heads. 

Borg et al. [15] found a pressure transfer to central metatarsophalangeal joints (second to fourth MH) after resection of the first metatarsal head and hallux amputation, similar to our results. Similarly, Patel et al. [14] observed a transference of pressure to the mid-region after resection of the second or third metatarsal heads, but there was no significant difference (*p* = 0.11). None of these studies found a displacement to the medial compartment of plantar pressure after resection of the fourth or fifth metatarsal heads, perhaps because of the small sample size of the studies compared to our sample.

The increase of pressure explains the reulceration in cases of resection of the first and second metatarsal heads. Surprisingly, there were patients in which the highest PPP and PTI after metatarsal head resection were not related to the development of a reulceration. These patients developed a reulceration in a different location. This could be explained in that we did not use in-shoe devices to screen the effect of the custom made in-soles. In addition, shear pressure was not analyzed, and this could be another factor affecting the ulcer occurrence [28].

Previous studies of our research group [25] demonstrated that there was a high probability of developing a reulceration event after an MHR of the first metatarsal head. After resection of the first metatarsal head, our results showed that the second metatarsal head suffered from the highest PPP and PTI, and it will develop a new ulcer in a year despite the use of orthotic interventions. A similar trend was found after resection of the second metatarsal head. The first metatarsal head will suffer irrevocably from a reulceration. Further studies could investigate the effect of prophylactic surgery to analyze its preventive power in these patients. 

We recommend separately analyzing plantar pressure in the different metatarsal heads instead of groups, as has been previously described [14,15]. This can establish preventive treatment in localized areas which are at high risk of DFU. Our results demonstrate a standardization of loading patterns after a head resection. This could help non-specialized clinicians screen foot biomechanical complications. The loading patterns can predict reulceration via a standardized protocol after head resection procedures.

However, our results should also be interpreted with caution due to the following limitations. Our population presented a high percentage of toe amputations and forefoot deformities, and it could influence the increase of barefoot pressures. Additionally, we did not measure the pressures before the intervention, and therefore we cannot measure with certainty that this was really the reason for the patterns we describe. Further studies could analyze these factors as potential risks. Nevertheless, this is the first prospective study in the literature to demonstrate a transfer pattern of PPP and PTI after MHR procedure—a common process used in routine diabetic foot surgery.

## 5. Conclusions

Patients who underwent a minor metatarsal head resection (second–fifth metatarsal heads) showed a medial transference of pressure. Additionally, following the resection of the first metatarsal head there was a transference of pressure beneath the 2nd metatarsal head. Increase of pressure was found to be a predictor of reulceration in cases of resection of the first and second metatarsal heads.

## Figures and Tables

**Figure 1 jcm-10-02260-f001:**
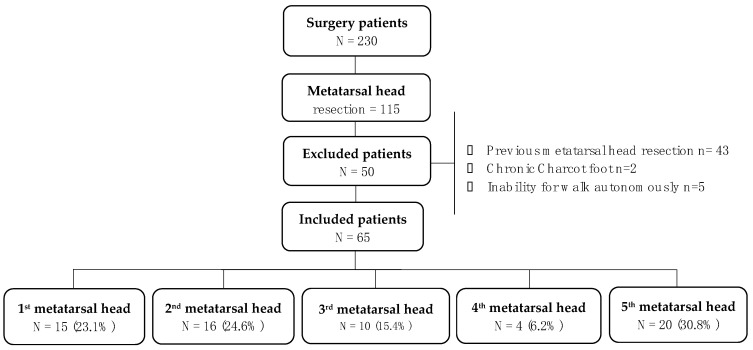
Study flow diagram. Legend: Flow chart of the 230 participants screened. After exclusion criteria 65 patients were included for analyses.

**Table 1 jcm-10-02260-t001:** Baseline clinical data.

Baseline Characteristics	Patients (*N* = 65)
Male/Female, *n* (%)	61 (93.8%)/4 (6.2%)
Type 1/Type 2 Diabetes, *n* (%)	15 (23.1%)/50 (76.9%)
Retinopathy, *n* (%)	30 (46.2%)
Hypertension, *n* (%)	55 (84.6%)
Nephropathy, *n* (%)	16 (24.6%)
Cardiopathy, *n* (%)	21 (32.3%)
Diabetic polyneuropathy, *n* (%)	65 (100%)
PAD, *n* (%)	23(35.4%)
Mean age ± SD (years)	64.28 ± 10.74
Body mass index (kg/m^2^), mean ± SD	28.27 ± 6.0
Diabetes mellitus (years), mean ± SD	23.06 ±11.38
Glycated hemoglobin (%), mean ± SD	7.78 ± 1.42
**Foot Characteristics**	
Previous partial toe amputation, *n* (%)	40 (61.5%)
Hallux abductus valgus, *n* (%)	6 (9.2%)
Taylor´s bunion, *n* (%)	14 (21.5%)
Hammer toe, *n* (%)	47 (72.3%)
Metatarsal prominence, *n* (%)	49 (75.4%)
**Baseline pressure distribution**	
PPP forefoot area (N/cm^2^) (1st to 5th metatarsal heads) PTI forefoot area (N/cm^2^/s) (1st to 5th metatarsal heads)	72.92 ± 26.94 28.77 ± 13.26

Legend: DM, diabetes mellitus; PAD, peripheral arterial disease; PPP, peak plantar pressure; PTI, pressure time integral; MH, metatarsal head; SD, standard deviation.

**Table 2 jcm-10-02260-t002:** Peak plantar pressure in the adjacent metatarsal heads after the first metatarsal head resection procedure.

Metatarsal Head Resected	Peak Plantar Pressure N/cm^2^					*p* Value (95% CI)
	1st Metatarsal	2nd Metatarsal	3rd Metatarsal	4th Metatarsal	5th Metatarsal	
1st MHR	-	22.53 [31.20–16.70] *	12.14 [16.26–8.26]	9.46 [21.60–2.93]	6.66 [10.10–0.60]	<0.001 * (18.01–31.42)
2nd MHR	25.93 [31.57–14.99] *	-	13.40 [25.42–7.33]	9.0 [17.77–5.16]	5.46 [11.50–2.35]	0.001 * (17.41–31.66)
3rd MHR	26.42 [29.14–22.57] *	16.65 [27.28–8.70]	-	14.30 [15.96–12.95]	5.08 [15.62–1.51]	0.001 * (22–32.83)
4th MHR	8.93 [14–4.81]	27.21 [30.66–24.53] *	25.93 [27.46–18.20]	-	16.10 [18.81–11.70]	0.009 * (22.40–32.54)
5th MHR	10.13 [17.10–5.76]	19.10 [22.89–10.00]	23.19 [25.96–18.95] *	14.65 [16.91–13.09]	-	<0.001 * (20.59–24.86)

Legend: MHR, metatarsal head resection; N/cm^2^, Newton/centimeter^2^. * Differences were assumed significant at *p* < 0.05 for a confident interval of 95%.

**Table 3 jcm-10-02260-t003:** Pressure time integral in the adjacent metatarsal heads after the first metatarsal head resection procedure.

Metatarsal Head Resected	Pressure Time Integral N/cm^2^/s					*p* Value (95% CI)
	1st Metatarsal	2nd Metatarsal	3rd Metatarsal	4th Metatarsal	5th Metatarsal	
1st MHR	**-**	6.73 [12.30–1.93] *	6.33 [12.30–1.93]	3.13 [5.86–0.96]	2.16 [3.86–0.16]	0.003 * (3.95–10.06)
2nd MHR	8.75 [14.56–3.70] *	**-**	5.06 [10.69–3.43]	3.41 [6.75–2.57]	1.85 [6.87–1.04]	0.02 * (5.90–13.13)
3rd MHR	11.86 [14.86–9.26] *	4.51 [8.07–2.14]	**-**	6.82 [7.70–5.90]	1.96 [6.65–0.37]	<0.001 * (8.81–18.33)
4th MHR	2.10 [4.96–0.75]	14.25 [16.45–12.65] *	7.83 [12.80–1.65]	**-**	10.93 [14.54–4.60]	0.02 * (11.26–17.63)
5th MHR	3.41 [4.75–1.44]	4.76 [6.45–2.26]	11.33 [13.15–9.67] *	5.00 [7.97–2.90]	**-**	<0.001 * (10.2–12.39)

Legend: MHR, metatarsal head resection; N/cm^2^/s, Newton/squared centimeters/seconds.* Differences were assumed significant at *p* < 0.05 for a confident interval of 95%.

## Data Availability

The data are available previous request to corresponding author.

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
