# Peer review of "Analysis of Plantar Pressure Pattern after Metatarsal Head Resection. Can Plantar Pressure Predict Diabetic Foot Reulceration?"

_jcm, 2021, doi:10.3390/jcm10112260_

Round 1

Reviewer 1 Report

The statistical methodology is inadequate and the low sample size does not allow for comparison between groups.

Author Response

The statistical methodology is inadequate, and the low sample size does not allow for comparison between groups.

Thank you very much for your comment.

Regarding the sample size calculation, it was previously calculated in the study protocol of which this work is part, and this section has been added in the manuscript accordingly:

“The target sample size was calculated using Epidat® v.4.2 for Mac OS. It was determined that the standard deviation was 5.4 N/cm2 to detect a difference of at least 2 times for peak plantar pressure changes after metatarsal head resection [14], based on a desired power of 80% with a β level of 20%, α level of 0.05, and confidence interval of 95%. Assuming a loss of 0.1% due to observational study design, at least 64 participants must be included in the study”

It is probably that we cannot determine the power of confounding factors on the transfer of peak plantar pressure after a metatarsal head resection. Additionally, as another reviewer comments, we did not measure the pressures before the intervention, and therefore we cannot measure with certainty that this was really the reason for the patterns. However, this, it is understood that there is a statistical association that has showed differences between groups, therefore, we can assume that the causal connection exists. Nevertheless, this part has been assumed as a limitation of the work in the discussion section.

For the primary outcome analyses, we tried to explore the metatarsal that supports the highest plantar pressure after a metatarsal head resection. We have employed Friedman test for paired samples, because of the non-normal distribution of the variable. It was a typographical error in the manuscript, and it was changed accordingly. We talk about an independent test for non-parametric samples, and we mentioned Friedman, the reviewer is probably referring to this. It was modified accordingly.

Additionally, for the secondary outcome analyses, we tried to explore the relation among the highest PPP and PTI with reulceration using Wilcoxon-Mann Whitney for independent samples because of the non-parametric distribution of the sample.

Reviewer 2 Report

The manuscript entitled “Analysis of plantar pressure pattern after metatarsal head resection. Can plantar pressure predict diabetic foot reulceration ?.” although a very interesting study. Thanks to the results it provides, we can make a preventive approach to the appearance of ulcers in patients with MD.

But I recommend that the authors address these considerations before being published. publication.

-In the methodology section, the authors should make an explanation about the manufacture of the plantar supports more focused on the biomechanical, morphological and structural requirements that were pursued.

- figure 1 is not sharp enough to be published.

- The authors must reflect as a limitation that they did not measure the pressures before the intervention, and therefore cannot measure with certainty that this was really the reason for the patterns they describe. I also invite the authors to reflect on this issue in the discussion.

Author Response

The manuscript entitled “Analysis of plantar pressure pattern after metatarsal head resection. Can plantar pressure predict diabetic foot reulceration ?.” although a very interesting study. Thanks to the results it provides, we can make a preventive approach to the appearance of ulcers in patients with MD. But I recommend that the authors address these considerations before being published. publication.

-In the methodology section, the authors should make an explanation about the manufacture of the plantar supports more focused on the biomechanical, morphological and structural requirements that were pursued.

Thank you for your comment. The biomechanical assessment in the methodology section was modified accordingly. The following description has been added:

“After biomechanical assessment, the same experienced podiatrist prescribed a custom-made insole. A positive plaster cast of the foot was created from a static foam box impression of the foot under semi-weight bearing position. The metatarsal region on the plaster cast was marked to guide the offloading (cut-out and metatarsal bar). The insole consisted of a 5 mm micro-cork base added to a 6 mm mid-layer of ethylene vinyl acetate (EVA, shore 40A). Apex height of the metatarsal bar was 8-10mm. Additionally, in the areas identified at risk because of the highest PPP and PTI, 5 mm EVA was removed and padded with 3 mm Plastazote®, shore 25A. Finally, the insoles were finished with a top cover of 3-mm-thick PPT® [22]. In addition, the subject was fitted with an extra-depth therapeutic shoe consisting of a rigid rocker outsole with a pivot point at 60 percent of shoe length, a rocker angle of 20 degrees, and an upper of stretch material (Podartis, Montebelluna, Italy) [23].”

- figure 1 is not sharp enough to be published.

Thank you for your appreciation. Figure 1 has been modified, we changed the font size and we have highlighted the statements in bold to make it clearer.

- The authors must reflect as a limitation that they did not measure the pressures before the intervention, and therefore cannot measure with certainty that this was really the reason for the patterns they describe. I also invite the authors to reflect on this issue in the discussion.

Thank you for your recommendation. It has been added as a limitation in the discussion section:

“Additionally, we did not measure the pressures before the intervention, and therefore we cannot measure with certain that this was really the reason for the patterns we describe.”

Reviewer 3 Report

This is a suffuciently cleraly written paper, is interesting for readership of this specific specailty. I have only a couple of comments:

  1. the number of patients is low and this may raise concrerns about statistical power iof the sample.
  2. This does not permit any further multivariate statistics for possible confouders such as for instance body weight or metabolic control
  3. What kind of previous amputation in Table 1?
  4. Authors should better detail and hightline main messages from ths dat in Conclusion and in Abstract

Author Response

This is a suffuciently cleraly written paper, is interesting for readership of this specific specailty. I have only a couple of comments:

  • The number of patients is low and this may raise concerns about statistical power iof the sample.

The sample size was previously calculated in the study protocol of which this work is part, and this section has been added in the manuscript accordingly:

“The target sample size was calculated using Epidat® v.4.2 for Mac OS. It was determined that the standard deviation was 5.4 N/cm2 to detect a difference of at least 2 times for peak plantar pressure changes after metatarsal head resection [14], based on a desired power of 80% with a β level of 20%, α level of 0.05, and confidence interval of 95%. Assuming a loss of 0.1% due to observational study design, at least 64 participants must be included in the study”

For that reason, power of the sample was not calculated because we reached the 100% sample size calculation.

  • This does not permit any further multivariate statistics for possible confouders such as for instance body weight or metabolic control

Thank you for your comment, we greatly appreciate it.

As you mentioned, this does not permit a multivariate analysis. However, although we are not able to know the influence of confounding factors on the transfer of peak plantar pressure after a metatarsal head resection, it is understood that there is a statistical association that has showed differences between groups, therefore, we can assume that the causal connection exists.

  • What kind of previous amputation in Table 1?

“Previous amputation” in Table 1, concerns to any amputation that involves the toes and that could involve the following conservative procedures: partial distal phalangectomy, distal Syme amputation, interphalangeal joint (IPJ) arthroplasty.

It has been modified in the table accordingly “Previous partial toe amputation”.

Lázaro-Martínez JL,García-Madrid M, García-Álvarez, Álvaro-Afonso FJ, Sanz-Corbalan I, García-Morales E. Conservative surgery for chronic diabetic foot osteomyelitis: Procedures and recommendations. J Clin Orthop Trauma. 2020;16:86-98.doi: 10.1016/j.jcot.2020.12.014.

  • Authors should better detail and hightline main messages from ths dat in Conclusion and in Abstract

Thank you very much for your recommendation.

The conclusion and the abstract have been reworded to highlight the main message:

“Patients who underwent a minor metatarsal head resection (2nd – 5th metatarsal heads) showed a medial transference of pressure. Additionally, following the resection of the first metatarsal head there was a transference of pressure beneath the 2nd metatarsal head. Increase of pressure was found to be a predictor of reulceration in cases of resection of the 1st and 2nd metatarsal heads.”

Round 2

Reviewer 1 Report

Dear Authors 
in the methods you have indicated that the multiple comparisons have been carried out with the Friedman test, why in table 2 you report the significance of the test with the confidence intervals of the 95% estimates and you do not indicate the result of this the test ?

Author Response

Dear authors in the methods you have indicated that the multiple comparisons have been carried out with the Friedman test, why in table 2 you report the significance of the test with the confidence intervals of the 95% estimates and you do not indicate the result of this the test?

 Thank you very much for your appreciation.

As you mention, we report the significance of the test with p values < 0.05 that are considered statistically significant with a confidence interval of 95%. Following your recommendation, the confidence intervals has been added in Table 2 and 3.